# Detection of Hypoglycemia Using Measures of EEG Complexity in Type 1 Diabetes Patients

**DOI:** 10.3390/e22010081

**Published:** 2020-01-09

**Authors:** Maria Rubega, Fabio Scarpa, Debora Teodori, Anne-Sophie Sejling, Christian S. Frandsen, Giovanni Sparacino

**Affiliations:** 1Department of Information Engineering, University of Padova, 35131 Padova, Italy; maria.rubega@unipd.it (M.R.); fabio.scarpa@dei.unipd.it (F.S.); debora.teodori@studenti.unipd.it (D.T.); 2Department of Neurosciences, University of Padova, 35128 Padova, Italy; 3Department of Nephrology and Endocrinology, Nordsjællands Hospital, 3400 Hillerød, Denmark; Anne-Sophie.Sejling@regionh.dk (A.-S.S.); frandsenmail@gmail.com (C.S.F.); 4Department of Obstetrics and Gynaecology, Nordsjællands Hospital, 3400 Hillerød, Denmark

**Keywords:** entropy, complexity measures, time-series analysis, EEG, type 1 diabetes, hypoglycemia, neural network classification

## Abstract

Previous literature has demonstrated that hypoglycemic events in patients with type 1 diabetes (T1D) are associated with measurable scalp electroencephalography (EEG) changes in power spectral density. In the present study, we used a dataset of 19-channel scalp EEG recordings in 34 patients with T1D who underwent a hyperinsulinemic–hypoglycemic clamp study. We found that hypoglycemic events are also characterized by EEG complexity changes that are quantifiable at the single-channel level through empirical conditional and permutation entropy and fractal dimension indices, i.e., the Higuchi index, residuals, and tortuosity. Moreover, we demonstrated that the EEG complexity indices computed in parallel in more than one channel can be used as the input for a neural network aimed at identifying hypoglycemia and euglycemia. The accuracy was about 90%, suggesting that nonlinear indices applied to EEG signals might be useful in revealing hypoglycemic events from EEG recordings in patients with T1D.

## 1. Introduction

Although type 1 diabetes (T1D) is one of the most common endocrine and metabolic conditions, little is known about brain dysfunction during hypoglycemia (i.e., a blood glucose (BG) concentration of <70 mg/dL (3.9 mmol/L)) [1]. The human brain depends on a continuous supply of glucose, which is the main source of energy. [2] Consequently, blood glucose regulation is critical for the brain’s physiology, and the brain is vulnerable to any glucose deprivation. Despite the brain accounting for 2% of body weight, it is the principal consumer of glucose (i.e., 20% of glucose-derived energy) [3]. The brain needs glucose for neuronal and non-neuronal cellular maintenance and for the generation of neurotransmitters [2]. Furthermore, glucose is required for both neurotransmitter synthesis and several glucose-metabolizing enzymes controlling cellular survival. However, the largest amount of glucose is consumed for neural computation and signal processing. During activation, the brain increases its glucose consumption [4]. Thus, when BG levels are too low, brain functions become impaired and the ionic currents within neurons of the brain are affected, leading to both short-term and possibly also long-term cognitive dysfunction [5]. For instance, in people with T1D, it has been shown that early exposure to severe hypoglycemia may have lasting and clinically relevant effects on cognition [6]. In general, hypoglycemia results in altered cerebral activity that is measurable in the voltage fluctuations of an electroencephalography (EEG) signal [7].

Previous studies have investigated whether cognitive dysfunction alters EEG activity during hypoglycemia. In the 1940s, it was observed that abnormal EEG activity (characterized by an increase of delta and theta activity and irregular or disorganized patterns) occurs during hypoglycemia. Later, researchers emphasized the importance of BG concentration on cerebral function, observing EEG dysrhythmia (i.e., distorted EEG waves with higher amplitude) during hypoglycemia [8,9]. Moreover, there is some evidence showing that power in low-frequency EEG bands (<8 Hz) increases during hypoglycemia [10,11], while a decrease in EEG scalp connectivity was seen in the path from the occipital lobes to the temporal and central lobes during hypoglycemia in the theta (O1→C4, O2→Cz, T5→C3) and alpha (O1→T4, O1→C4) bands by computing the directional effect of one EEG recording channel over another through the information partial directed coherence function [12,13,14]. Recently, multi-scale entropy algorithms based on the computation of Sample Entropy (SampEn) found a decrease in EEG complexity among T1D patients during a hyperinsulinemic–hypoglycemic clamp [15]. Although the use of nonlinear indices might open new insight into the analysis and understanding of the brain’s functions in hypoglycemia, the high computational cost of SampEn limits its use for real-time applications (e.g., for hypoglycemia detection). Other entropy-based methods, such as empirical permutation entropy (ePE) and empirical conditional entropy (eCE), are able to overcome the limitations of SampEn [16]. We showed (on a limited database) that fractal dimension indices (e.g., the Higuchi index [17], residuals, and tortuosity [18]) can be effectively and efficiently used in the detection of hypoglycemic events from EEG signals. Indeed, fractal dimension indices, besides having no need for storage memory, are classified as linear time algorithms (i.e., they have a time complex O(N), such as ePE and eCE), whereas multi-scale entropy methods (e.g., SampEn) are classified as quadratic time algorithms (i.e., they have a complex O(N^2^)). Proving the sensitivity to hypoglycemia and the computational efficiency of scalp EEG complexity indices are pivotal steps in developing future automatic systems that are able to use the brain as a biosensor to detect hypoglycemia under approximately real-time and daily-life conditions. Thus, the aim of this study is (1) to investigate scalp EEG dynamics changes by ePE, eCE, and fractal dimension approaches on an updated and extended EEG database previously presented in [15,16] and (2) to develop a neural network (NN) classifier to identify hypoglycemic events, starting from a selection of nonlinear indices extracted from multiple EEG channels that alone or in combination are the most sensitive to hypoglycemia.

## 2. Materials and Methods

### 2.1. Database and Experimental Protocol

Data were partially taken from [6]. This study was approved by the local ethical committee and registered at clinicaltrials.gov with identifier number NCT01337362 (18 April 2011). All subjects gave their informed consent for inclusion before they participated in the study. We selected 34 patients with T1D (14 hypoglycemia-aware; 19 males; mean (±standard deviation) age: 55 ± 3 years; diabetes duration: 28 ± 3 years). The inclusion and exclusion criteria have been described in detail in [6]. Briefly, these criteria were T1D for >3 years, with an age >18 years. Exclusion criteria included pregnancy; breastfeeding; any brain disorder; use of antiepileptic drugs, β-blocking drugs, or neuroleptic drugs; use of benzodiazepines within the last months; cardiovascular disease; and alcohol or drug abuse. Each patient underwent an 8 h hyperinsulinemic–hypoglycemic clamp procedure with intermittent neuropsychological evaluation. For a target of 5.0–6.0 mmol/L (euglycemia), 2.0–2.5 mmol/L (euglycemia), and finally 5.0–6.0 mmol/L again (recovery), the patients were intravenously administered Actrapid insulin (Novo Nordisk, Ballerup, Denmark) mixed with their own heparinized plasma and isotonic saline, at a rate of 1 mU insulin/kg/min, in order to progressively induce a hypoglycemic state. To keep the plasma glucose at the desired level, glucose (20%) was administered at a variable rate. During the whole period, BG samples (Yellow Springs Inc., Yellow Springs, OH, USA) were monitored every 5 min to obtain at least one hour of hypoglycemia (BG < 70 mg/dL) and one hour of euglycemia (BG in 70–180 mg/dL). Simultaneously, 21 EEG channels were recorded (Cadwell, Easy II, Kennewick, WA, USA) using standard cap electrodes placed on the scalp according to the international 10/20 system, while the patients were sitting in a chair with their eyes open. Offline, the EEG was analogically low-pass filtered to avoid aliasing, and then the signal was digitally acquired. During each glycemic state (i.e., euglycemia and hypoglycemia), 5 min EEG signals under resting conditions were selected from the BG time series (smoothed by a spline approximation of the samples). We took 5 min EEGs, discarding all the ascending and descending phases between hypoglycemia and euglycemia (which had different durations among the subjects) and discarding the EEG epochs characterized by eye and muscle artifacts to minimize the pre-processing and manipulation of the EEG signal before analysis, particularly in the frontal regions.

### 2.2. EEG Pre-Processing

EEG data for euglycemia and hypoglycemia were zero-phase digitally filtered between 1 and 40 Hz using a four-order Butterworth filter; independent component analysis (ICA) was applied in order to remove cardiac and oculomotor artifacts (only components with clear eye blinks, saccades, and cardiac artefacts were excluded). Artifact stereotypical ICs were determined by evaluating their topographies, time courses, and power spectra (1 to 7 ICs per subject were rejected). After ICA correction, EEG signals exceeding amplitudes of ±200 μV were excluded in the subsequent analysis (80% of the raw data remained for further analysis). EEG signals were re-referenced to the common average reference.

Eventually, from each 5 min interval, 4 s EEG epochs (i.e., non-overlapping segments of 800 samples (sampling frequency = 200 Hz)) were extracted for each patient. The nonlinear indices were computed for each of these 4 s EEG epochs.

In Figure 1, the P3-A1A2 EEG channel recording simultaneous BG data from a representative patient are presented over the whole 8 h of acquisition. In the first hour, glucose levels increased until the insulin preparation used in the clamp procedure was ready, because patients were instructed to not take their morning insulin on the day of the study. Then, at least one hour of euglycemia (referred to as EU in the figures), one hour of hypoglycemia (HYPO), and one hour of recovery after induced hypoglycemia were identified by visual inspection of the BG time series (these time intervals are highlighted by the green and blue semi-transparent areas in Figure 1).

A qualitative inspection of the EEG recording in Figure 1 illustrates the generalized slowing of cerebral activity and increase in regularity described in the literature [15,16].

### 2.3. Assessment of EEG Complexity in Euglycemia and Hypoglycemia

#### 2.3.1. Entropy Measures

Permutation entropy indices are measures that are free of restrictive parametric model assumptions and robust to noise. These indices account for the temporal ordering structure of a given time series of real values, allowing the user to unlock the complex dynamic content of a nonlinear time series [20,21]. They have been proven to have a lower computational cost, to be robust to strictly monotone transformations (such as transformations caused by changing the acquisition system), and less dependent on the length of a time series.

In particular, the ePE of order d ∈ ℕ and the ePE of delay *τ* ∈ ℕ of a time series (*x_t_*${)}_{t=0}^{N-1}$ with N ∈ ℕ is given by
(1)$$\mathrm{ePE}(d,\;\tau ,\;(x_{t}{)}_{t=0}^{N-1})\;=\;-\frac{1}{d}\sum_{\pi \in \Pi_{d}}p_{\pi }^{\tau }\ln\left(p_{\pi }^{\tau }\right),$$
where
(2)$$p_{\pi }^{\tau }\;=\;\frac{\#\{\;t\in \left\{d\tau ,\;d\tau +1,\ldots ,\;N-1\right\}\;|\;\left(x_{t},\;x_{t-\tau },\ldots ,x_{t-d\tau }\right)\;has\;ordinal\;pattern\;\pi \}}{N\;-\;d\tau -1}$$
is the relative frequency of ordinal patterns *π* in the time series, and 0 ln 0 is defined by 0. The vectors $\left(x_{t},\;x_{t-\tau },\ldots ,x_{t-d\tau }\right)$ related to *t*, *d*, and *τ* are called delay vectors. For instance, ePE was successfully applied in the previous biomedical signal processing literature to detect EEG changes related to epileptic seizures [22], to distinguish brain states related to anesthesia [23] or sleep stages [24], and to analyze and classify heart rate variability data [25].

The eCE for the ordinal patterns of order *d* ∈ ℕ and of delay *τ* ∈ ℕ of a time series (*x_t_*${)}_{t=0}^{N-1}$ with *N* ∈ ℕ is defined as
(3)$$\mathrm{eCE}(d,\tau ,\;\left(x_{t}{)}_{t=0}^{N-1}\right)\;=\;\sum_{\pi \in \Pi_{d}}p_{\pi }^{\tau }\ln\left(p_{\pi }^{\tau }\right)-\sum_{\pi_{1},\pi_{2}\in \Pi_{d}}p_{\pi_{1},\pi_{2}}^{\tau }\ln\left(p_{\pi_{1},\pi_{2}}^{\tau }\right),$$
where $p_{\pi }^{\tau }$ is defined as in Equation (2) and
(4)$$p_{\pi_{1},\pi_{2}}^{\tau }\;=\;\frac{\#\{\;t\in \left\{d\tau ,\;d\tau +1,\ldots ,\;N-1\right\}\;|\;\left(x_{t},\;x_{t-\tau },\ldots ,x_{t-d\tau }\right)\;has\;ordinal\;pattern\;\pi_{1},\pi_{2}\}}{N\;-\;d\tau -1}.$$

eCE has been shown to have better performance than classical permutation entropy in some situations [26]. Thus, we also decided to test this ordinal measure with the final aim of obtaining the most accurate classification of the glycemic state through EEG data. We refer the reader to [26] for further details on permutation entropy measures and to [27] for further details on conditional entropy measures applied to EEG data.

As the final aim is to use ePE and eCE as the inputs for a classifier to detect hypoglycemia, the choice of the parameters is of fundamental importance, whereas the choice of an higher order *d* does not make sense for statistical reasons [26]. Testing different delays *τ* is useful, since the delay contains important scale information.

#### 2.3.2. Fractal Dimension Measures

The fractal dimension provides a complexity index that describes how the measure of the length of a curve *L*(*k*) changes depending on the scale *k* used as a unit of measurement. Different approximation techniques can be used to estimate the fractal dimension [28]. Among these different approaches, Higuchi’s algorithm [29] is often employed in EEG analysis to estimate the fractal dimension *D* [17,18,30]. Higuchi’s standard approach calculates the fractal dimension of a time series in the time domain and is based on the log[*L*(*k*)] versus log(*k*) curve computed as follows. (a) For each sample i of the EEG epoch s = {s(1), s(2), …, s(N)} of length *N*, the absolute differences between values s(*i*) and s(*i−k*) (i.e., the samples at distance *k*) are computed, considering *k* = 1, …, *kmax*; (b) then, each absolute difference is multiplied by a normalization coefficient that takes into account the different numbers of samples available for each value of *k*. The computation of this coefficient is based on the starting point *m* = 1, …, *k* and on the total number (*N*) of samples of an epoch; (c) *L*_m_(*k*) is computed as
(5)$$L_{m}\left(k\right)=\frac{1}{k}\cdot \left[\sum_{i=1}^{q}\left|s\left(m+i\cdot k\right)-s\left(m+\left(i-1\right)\cdot k\right)\right|\right]\cdot \frac{N-1}{q\cdot k},$$
where *q* = int[(*N-m*)/*k*], and *L*(*k*) is computed by summing the obtained values and dividing by *k*:(6)$$L\left(k\right)=\frac{1}{k}\cdot \sum_{m=1}^{k}L_{m}\left(k\right).$$

Here, the log[*L*(*k*)] versus log(*k*) curve, referred to in the following as *ℓk*, is finally derived. By definition, if *L*(*k*) is proportional to *k*^−*D*^ for *k* = 1,…, *klin*, where *klin* defines the upper range of the linear region of *ℓk*, then the curve is fractal with dimension *D*. Therefore, Higuchi’s fractal dimension *D* increases as the signal irregularity increases. The nonlinear part of the *ℓk* curve (*k* > *klin*) presents an oscillatory behavior whose characteristics depend on the periodicity of the signal itself [18]. Thus, we also extracted two additional features, which consider a wider domain of the *ℓk* curve with respect to the Higuchi fractal dimension [18]. The first feature (residuals) evaluates the deviation (i.e., the sum of the squares of the residuals) of the *ℓk* curve from the regression line computed in its linear region. The second feature is the tortuosity measures, $\tau_{\ell_{k}}$. This constitutes a measure of the rate at which the curve changes with respect to its coordinates changes (*x* = log(*k*) and *y* = log(*L*(*k*))) by using their first and second partial differences:(7)$$\tau_{\ell_{k}}=\sum_{n=3}^{k_{max}}\left|\frac{\Delta x\left(n\right)\Delta^{2}y\left(n\right)-\Delta^{2}x\left(n\right)\Delta y\left(n\right)}{{\left[{\left(\Delta x\left(n\right)\right)}^{2}+{\left(\Delta y\left(n\right)\right)}^{2}\right]}^{3/2}}\right|,$$
where Δ*x*(*n*) = *x*(*n*) − *x*(*n* − 1), Δ^2^*x*(*n*) = Δ*x*(*n*) − Δ*x*(*n* − 1), Δ*y*(*n*) = *y*(*n*) − *y*(*n* − 1), and Δ^2^*y*(*n*) = Δ*y*(*n*) − Δ*y*(*n* − 1). A complete description of the tortuosity measures can be found in [31]. These two features increase as the signal regularity and periodicity increase. We refer the reader to [18] for further details on EEG signal feature extraction based on the fractal dimension.

#### 2.3.3. Criteria for Statistical Analysis

To assess the significant differences for all the measures described in the previous paragraphs between hypoglycemia and euglycemia, paired Student’s t tests were computed for each channel and each patient, under the hypothesis of a normal distribution of samples (Lilliefors test). Otherwise, Wilcoxon rank-sign tests were considered (a value of *p* < 0.05 was considered significant).

### 2.4. Hypoglycemia Detection through Indices of EEG Complexity

To verify if nonlinear EEG indices can be used to detect hypoglycemia with good accuracy, we trained and tested a neural network (NN) on our EEG data to evaluate both individual measures and different combinations of complexity indices. We tried different values of the NN hyperparameters (e.g., the numbers of layers, number of neurons, type of activation function, learning rate, momentum, etc.). Eventually, we adopted the scheme of the NN architecture reported in Figure 2. As input for the NN, we used the values of the fractal dimensions (i.e., Higuchi, residuals, and tortuosity) and the ePE for each 4 s EEG epoch during both euglycemia and hypoglycemia in T5-A1A2, T6-A1A2, P3-A1A2, P4-A1A2, Pz-A1A2, O1-A1A2, and O2-A1A2. The data were randomly divided into two *N*-folds: one representing the training set and the other representing the testing set. The training of the NN and respective testing was repeated 1000 times, and each time 75% of the dataset was selected as a training set and the remaining 25% was selected as a testing set; these sets were always different for each analysis performed.

In order to evaluate the classification provided by the NN, we computed the accuracy, sensitivity, specificity, and precision of the classification by defining the following. For a true positive (TP), a 4 s EEG epoch during hypoglycemia was correctly classified; for a true negative (TN), a 4 s EEG epoch during euglycemia was correctly classified; for a false positive (FP), a 4 s EEG epoch during euglycemia was classified by the NN as a hypoglycemic state; for a false negative (FN), a 4 s EEG epoch during hypoglycemia was classified by the NN as a euglycemic state.

## 3. Results

### 3.1. Assessment of EEG Complexity in Euglycemia and Hypoglycemia

#### 3.1.1. Entropy Measures

To apply ePE and eCE to our EEG epochs, order *d* was set equal to 6—i.e., the maximum value for our window size was set according to the recommendation in [24], and different values of *τ* were tested (1–3). The reported results will refer to a *τ* set equal to 1, because this value maximized the difference between hypoglycemia and euglycemia in the selected EEG epochs. In Figure 3, both ePE and eCE provide a quantitative measure of the complexity of the signal in HYPO (blue) and EU (green) in subject #27 (randomly taken from the database (Figure 3, left panels)) and in all subjects (Figure 3, right panels). ePE and eCE capture the order relations between values of a time series and extract a probability distribution of the ordinal patterns. The smaller the ePE and eCE are, the more regular and more deterministic the time series. Contrarily, the closer to 1 the ePE and eCE are, the more noisy and random the time series. Indeed, an EEG signal for hypoglycemia provides more regular and more deterministic results than an EEG signal for euglycemia.

#### 3.1.2. Fractal Dimension Measures

For the EEG signal features based on the fractal dimension, the linear region was defined by imposing a klin equal to 6, according to [17], while for the residuals and the tortuosity, kmax was set as equal to 30, according to [18]. In Figure 4, the results for the fractal dimension are reported for the same subject as Figure 3 and also for the population of subjects. Despite the higher intersubject variability (Figure 4, right panels), when focusing on the single subject results (Figure 4, left panels), the fractal dimension Higuchi tends to decrease when passing from euglycemia to hypoglycemia, and the residuals and tortuosity tend to increase. Indeed, the decrease in the fractal dimension Higuchi denotes a decrease in the irregularity of the signal, whereas an increase in the residuals and tortuosity matches with an increase in signal regularity and periodicity. The fractal dimensions suggest that the EEG signal, for hypoglycemia, describes a process that is more regular and less complex.

The results of fractal dimension features (i.e., Higuchi, residuals, and tortuosity) for the same representative subjects (left panels) and for all subjects (right panels) are shown in Figure 3. On the left, boxplots are provided, representing the values of (a) Higuchi, (c) tortuosity, and (e) residuals computed from all the 4 s EEG epochs for subject #27 in EU (green) and HYPO (blue) glycemia, alongside topographic plots representing the median value for each EEG channel for EU and HYPO glycemia. On the right, boxplots are provided, representing the median values over all 4 s epochs of (b) Higuchi, (d) tortuosity, and (f) residuals for each subject in EU (green) and HYPO (blue) glycemia, as well as topographic plots representing the median value over all subjects for each EEG channel in EU and HYPO glycemia.

#### 3.1.3. Statistical Analysis

When analyzing each subject and each channel individually, the EEG signal presents (as reported for the representative subject in Figure 3 and Figure 4) statistically significant differences between euglycemic and hypoglycemic states in the majority of EEG channels, whereas a high intersubject variability in the median values of the complexity indices can be seen in Figure 3 and Figure 4. Despite this high intersubject variability in the median values of the complexity indices, we found that in the EEG signals recorded from at least two of the following channels—T5-A1A2, T6-A1A2, P3-A1A2, P4-A1A2, Pz-A1A2, O1-A1A2 and O2-A1A2, ePE, and the Higuchi fractal dimension—the residual and tortuosity features are always more statistically significantly different when passing from euglycemia to hypoglycemia in each subject. Tortuosity is the only feature that statistically significantly increases when passing from euglycemia to hypoglycemia in the majority of EEG channels (at least 15 over 19) in each subject. This more stable feature of the nonlinear region of the ℓk curve suggests a diffused increase in the signal regularity and periodicity during hypoglycemia.

Significant differences in discriminating between euglycemia and hypoglycemia in hypoglycemia-aware and hypoglycemia-unaware patients were not found in the three fractal dimension indices. On the other hand, evaluating ePE and eCE, the differences in the values between the two glycemic conditions were maximized in the hypoglycemia-aware subjects.

### 3.2. Hypoglycemia Detection through Indices of EEG Complexity

Classification of the glycemic state through fractal dimension measures (Higuchi, tortuosity, and residuals) and the ePE was deepened by applying a NN. This analysis will be useful to investigate if the information provided by each index is the same as that provided by the others, as well as to find the combination of indices that maximizes the ability to differentiate between glycemic states. Since the multi-channel analysis of EEG complexity features demonstrated a significant (*p* < 0.05) decrease in complexity during hypoglycemic episodes in either the temporal, parietal, or occipital brain regions in all subjects, we focused on the measures extracted by the following EEG channels: T5-A1A2, T6-A1A2, P3-A1A2, P4-A1A2, Pz-A1A2, O1-A1A2, and O2-A1A2 (Figure 3 and Figure 4). We discarded the frontal EEG channels in the following analysis because the EEG signal is usually massively corrupted by eye and muscle artifacts. Thus, these signals are highly dependent on pre-processing and IC selection.

To evaluate the classification provided by the NN, we computed the accuracy, sensitivity, specificity, and precision of the classification (Table 1). Considering the indices individually, the best classification results were obtained by the Higuchi index. However, satisfactory results were obtained only when considering the fractal indices together. The results slightly improved by adding ePE. These findings demonstrate that the proposed complexity indices can accurately classify glycemic states, and that each index provides information different from that of other indices.

The proposed nonlinear indices were proven to add additional and complementary information to the classical linear features, such as the power spectrum values in the canonical EEG frequency bands (delta (1–4) Hz, theta (4–8) Hz, alpha (8–13) Hz, and beta (13–30) Hz). Indeed, the accuracy reached when exploiting only the power spectrum features equalled 71%, whereas considering all the proposed complexity indices and power spectrum features achieved an accuracy equal to 90%.

We also tested how the length of the EEG epochs can affect the results and therefore, the performance of NN classification. When choosing EEG epochs <4 s, the accuracy tended to decrease (≤85%), whereas for epochs ≥4 s, the accuracy tended to stabilize around 89%. 4 s seems to be the most optimal trade-off between the level of accuracy achievable and the computation time for possible real-time application.

For sake of completeness, in Table 2, we also report the computational time and storage memory needed to compute the complexity measures cited in this work.

## 4. Discussion

It is well established that during hypoglycemia, an EEG is characterized by increased activity in the delta and theta bands [10,11]. Recently, evidence has demonstrated that hypoglycemia is also associated with changes in some EEG features measurable by nonlinear indicators [15,16].

The first aim of the present work was to deepen the analysis of EEG complexity during hypoglycemia already published in [16]. We showed that a scalp EEG signal recorded during hypoglycemia is characterized by a decrease in ePE, Higuchi’s fractal dimension, and an increase in both the residual and tortuosity fractal dimensions. This highlights that an EEG signal tends to be more regular, more uniform, and less complex during hypoglycemia compared to during euglycemia. The second aim of this work was to develop an algorithm that is capable of identifying hypoglycemic events from EEG recordings. For this goal, an NN classifier was identified from signals in multiple EEG channels that, alone or in combination, showed the most sensitivity to hypoglycemia. The obtained accuracy, using 21 complexity features as inputs for the NN, was almost 90%. This result seems remarkable, given that other recent works aimed at detecting nocturnal hypoglycemia by EEG spectral moments reached a sensitivity and specificity of 85% and 52%, respectively [32]. In addition, a classifier fed by the sole power spectrum features on our data led to an accuracy of 71% only. This result gives further evidence that nonlinear indicators, such as entropy-based indices and fractal dimensions, carry information complementary to that of linear methods.

Although, according to the proof-of-concept nature of the present work, the proposed fractal dimension features evaluated in a multi-channel framework seem to be promising tools, several points require/deserve further investigation. As far as the methodology is concerned, future studies should investigate the NN configuration and its physiological interpretation, aiming at the highest accuracy possible. With regard to this aspect, building up a complexity–entropy causality plane [33,34] to represent in space the complexity features of the EEG signals under the two glycemic conditions might help in the selection of classification features and in deepening the physiological interpretation of the results. As far as the physiological interpretation of the results is concerned, a limit of the present work is the low resolution of the EEG system, which allowed us to study EEG features only in the scalp domain. With regard to this aspect, a source localization study based on a high-density resolution EEG system (≥64 channels) might help to elucidate which brain regions are most affected by hypoglycemia and cause both a decrease in complexity and an increase in delta and theta power during scalp EEG [35]. Finally, for possible clinical applications, the use of an EEG signal, as an alternative or, more likely, complement to BG continuous sensors [36,37] has been suggested [38]. However, an EEG-based hypoglycemia detection device still needs to be tested in a clinical setting, and fundamental issues, such as sensitivity, specificity, reliability, resolution, and the influence of activity must be carefully addressed before demonstrating real-time usability.

## Figures and Tables

**Figure 1 entropy-22-00081-f001:**
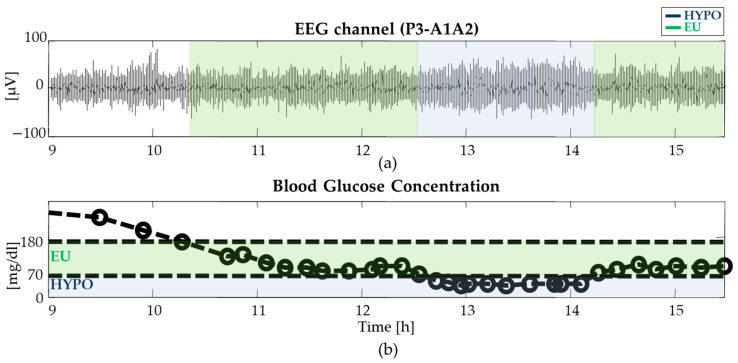
Data from a representative subject during the hyperinsulinemic–hypoglycemic clamp: (**a**) P3-A1A2 electroencephalography (EEG) recordings and (**b**) simultaneous blood glucose concentrations (open circles denote reference blood glucose samples and the dashed lines denote their smoothing spline interpolation [19]. Green and blue semi-transparent areas refer to euglycemia (EU) and hypoglycemic (HYPO) intervals, respectively.

**Figure 2 entropy-22-00081-f002:**
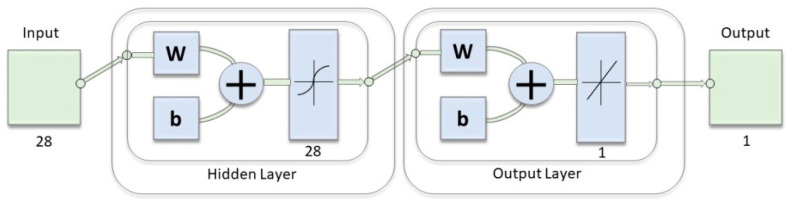
Graphic representation of the neural network (NN) implemented: a single hidden layer with a hyperbolic tangent sigmoid transfer function for the hidden and linear layer for the output neurons (neural network toolbox, MATLAB 2018b).

**Figure 3 entropy-22-00081-f003:**
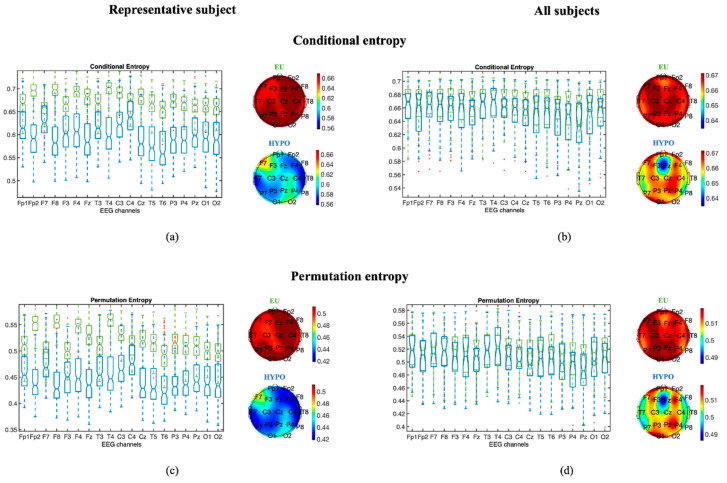
Results on a representative subject (left panels: (**a,c**)) and on all the subjects (right panels: (**b,d**)) for conditional entropy (top panels: (**a,b**)) and permutation entropy (bottom panels: (**c,d**)). Boxplots represent values of the two considered entropy indicators computed from all the 4 s EEG epochs in EU (green) and HYPO (blue). The two topographic plots reported on the right of each boxplot display the median values for each EEG channel, respectively, for EU and HYPO.

**Figure 4 entropy-22-00081-f004:**
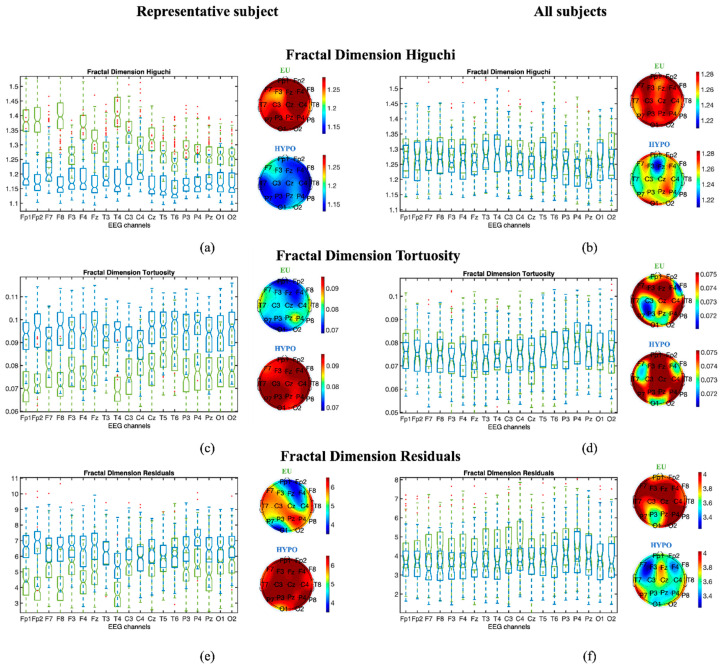
Results on a representative subject (left panels: (**a**,**c**,**e**)) and on all the subjects (right panels: (**b**,**d**,**f**)) for Higuchi (top panels: (**a**,**b**)), tortuosity (middle panels: (**c**,**d**)) and residuals (bottom panels: (**c**,**d**)). Boxplots represent values of the fractal indicators computed from all the 4 s EEG epochs in EU (green) and HYPO (blue). The topographic plots reported on the right of each boxplot display the median values for each EEG channel, respectively, for EU and HYPO.

**Table 1 entropy-22-00081-t001:** Classification results of the neural network classifier over 1000 repetitions using different combinations of EEG features. ePE: empirical permutation entropy.

	Accuracy		Sensitivity		Specificity		Precision	
	Testing Set	Training Set	Testing Set	Training Set	Testing Set	Training Set	Testing Set	Training Set
**All**	87.66 ± 0.80 ^1^	91.44 ± 0.67	86.81 ± 1.41	90.74 ± 1.06	88.51 ± 1.28	92.14 ± 0.86	88.33 ± 1.19	92.03 ± 0.80
**Fractal Dimension indices**	86.73 ± 0.86	89.91 ± 0.64	85.84 ± 1.52	89.10 ± 1.11	87.63 ± 1.37	90.72 ± 0.92	87.43 ± 1.27	90.57 ± 0.83
**Higuchi**	71.52 ± 1.46	72.50 ± 1.14	72.92 ± 3.66	73.92 ± 3.27	70.13 ± 3.24	71.08 ± 2.90	71.03 ± 2.01	71.93 ± 1.50
**Residuals**	68.44 ± 1.21	69.39 ± 0.83	63.71 ± 3.07	64.71 ± 2.64	73.18 ± 2.39	74.06 ± 1.97	70.43 ± 1.76	71.41 ± 1.05
**Tortuosity**	66.67 ± 1.13	67.73 ± 0.62	63.19 ± 3.15	64.24 ± 2.70	70.18 ± 2.88	71.21 ± 2.49	67.99 ± 1.83	69.10 ± 1.15
**ePE**	56.51 ± 0.98	56.79 ± 0.34	64.74 ± 3.33	64.96 ± 2.94	48.31 ± 3.08	48.61 ± 2.72	55.61 ± 1.33	55.84 ± 0.39

^1^ mean ± standard deviation in %.

**Table 2 entropy-22-00081-t002:** Computational time of the complexity measures.

	Computational Time	Storage
**SampEn**	O(*N*^2^)	O(*N*)
**ePE**	O(*N*)	O((*d* + 1)!(*d* + 1))
**eCE**	O(*N*)	O((*d* + 1)!(*d* + 1))
**Fractal dimension measures**	O(*kN*)	-

*N*: number of samples; *d*: entropy order; *k*: fractal dimension scale.

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
