# Peer review of "Detection of Hypoglycemia Using Measures of EEG Complexity in Type 1 Diabetes Patients"

_entropy, 2020, doi:10.3390/e22010081_

Round 1

Reviewer 1 Report

This paper considers several EEG entropy and fractal dimension measures as features to classify EU and HYPO states for T1D patients, with a neural network framework also applied. I read this manuscript with interest and provide the comments and suggestions below.

1) In introduction paragraph 2, it’s a bit confusing as “slow potential” and “fast potential” seems to be the two distinct descriptions.

2) In the introduction paragraph 2, best also specify how BG concentration differs on cerebral function.

3) In introduction paragraph 2, “…that power in the low frequency EEG…”, please specify the frequency band of this low frequency.

4) In introduction paragraph 2, “while EEG coherence in the posterior…”, EEG coherence between what? The averaged of each pair of electrode contacts in posterior area?

5) In introduction paragraph 2, since the authors pointed out the computational cost of MSE and then introduce the other methods such as ePE, eCE, and some other fractal dimension indices afterward, I would suggest specifying how these methods can improve computation and thus achieve real-time computation (e.g., specify required samples or window, % of improvement). I know the author discusses the computational time later in the result section, however, it would be good if briefly point this in the introduction.

6) In the database and experimental protocol section, what is the spec of the low-pass filter? Why only take 5-min EEGs for additional analyses as you have at least one-hour recording? Please also specify parameters in smoothing and spline approximation.

7) q-complexity-entropy curve, which is also based on permutation entropy, would be a suitable tool to visualize and optimize the features for NN. I would suggest the authors discuss this point.

Yeh et al., A novel method of visualizing q-complexity-entropy curve in the multiscale fashion, nonlinear dynamics, 2019.

Ribeiro et al., Complexity-entropy causality plane: a useful approach for distinguishing songs, physica a, 2012.

8) I wonder how the length of epochs can affect the EEG complexity and therefore the performance of NN classification?

9) I wonder how spectral power features could be used in NN classification as compared to those with EEG complexity since the real-time computation of spectral power is also feasible, it would be good if the authors could either discuss and compare with the performances in literature or provide the relevant results in your datasets.

10) In figure 3, the differences between HYPO and EU is quite small as compared to that of the representative subject, and thus we can infer that there could have some subject present higher entropy measure in HYPO than EU, which is opposite to that of the hypothesis, so among all #27 subjects, how many subjects show higher entropy measure in EU? And please discuss this point (also for figure 4).

Reviewer 2 Report

Major points

Section 2.2: Examples of the pre-processing of EEG to remove cardiac and oculomotor artifacts to be included together with more information on the pre-processing methods used. Lines 239-242: From Figure 4 right panels, I do not see the channels the authors claim are mostly affected. It appears that the frontal channels are mostly affected. There no apparent statistically significant differences in the right panels in Figures 3 and 4, that is, when all subjects are included.

Minor points

For the benefit of accuracy, a general point is that “EEG” should be replaced by “scalp EEG”. For example in the Abstract: “By using a dataset of 19 channel scalp EEG recordings…” Line 49: “….empirical Permutation entropy (ePE) and empirical Conditional Entropy (eCE)…” Line 54: “Proving the sensitivity and specificity to hypoglycemia and …” Figure 1. The vertical lines should be replaced by guidance from subfigure (b). As is, they seem to be subjectively drawn under the limitation of 1 hour segments. I suggest that the yellow and pink colors from (b) to be drawn in (a) exactly correspondingly. In this way, both yellow and pink color zones will be much bigger in (a). Line 105: “A qualitative inspection….”. (Omit “Even” as it cannot be apparent from the EEG in Figure 1 at the resolution provided) Line 150: kiln is not defined Line 156: “…It constitutes a measure of the rate…” Line 159: D2x(n) should be: D2x(n) etc. like in Eq. 7 Line 177-178: Delete “All available data were …training set” as it conveys the wrong impression. The following text specifies well how you treated the data (training and testing) Figure 3 labels: Include eCE and ePE like: Conditional Entropy (eCE) etc. Also, in the caption of the Figure, include what each color means. Line 213: ‘….of conditional entropy (b) and permutation entropy (d) for each…” Line 223: ‘Indeed, the decrease in the fractal dimension…whereas an increase in Residuals…” Figure 4 caption: include what each color means Line 230 and 234: “…(a) Higuchi…” Page 9 basically needs to be rewritten

Round 2

Reviewer 1 Report

The authors already addressed all my raised points, good luck.

Author Response

We thank the Reviewer 1. To improve English in this revised version, we sent the manuscript to the MDPI English editing service: All the changes referred to English language are highlighted in green in the manuscript.

Reviewer 2 Report

I would like to recognize the authors for trying to answer my questions. However:

FIRST. English has to improve throughout the manuscript. For example (excerpt from the authors' responses to my questions with my suggested edits):

"(lines 103-106 page 3) Artifact stereotypical ICs were selected determined by evaluating their topographies, time course and power spectra (from 1 to 7 ICs per subject were rejected). After ICA correction, EEG signals exceeding amplitudes of ?200 ?V were excluded in the ignored for subsequent analysis (80% of the raw data points remained in total for further analysis). EEG signals were re-referenced to the common average-reference.

(lines 257-264 page 8) Despite the higher inter-subject variability in the median values of the complexity indices, in the EEG signals recorded from the following channels: T5-A1A2, T6-A1A2, P3-A1A2, P4-A1A2, Pz-A1A2, O1-A1A2 and O2-A1A2, ePE, Higuchi fractal dimension, Residuals and Tortuosity features ARE statistically significantLY different in euglycemia than hypoglycemia in all subjects. Tortuosity is the only feature that statistically significantly increases passing from euglycemia to hypoglycemia in all EEG channels for every subject. This more stable feature of the ...suggests a diffused increase of the signal regularity and periodicity during hypoglycemia across all subjects."

"(Discussion section - lines 302-334 pages 9-10) It is well established that during hypoglycemia the EEG is characterized by increased activity in the delta and theta bands [10] [11]. Recently, evidence has been provided that hypoglycemia is also associated to changes in some EEG features measurable by nonlinear indicators [15] [16]. The first aim of the present work was to further the analysis of EEG complexity during hypoglycemia already published in [16]. We showed that the scalp EEG signal recorded during hypoglycemia is characterized by a decrease of in ePE and Higuchi's fractal dimension, and an increase in both Residuals....The second aim of the work was to develop an algorithm capable to identify hypoglycemic events from EEG recordings... complementary to that of linear methods.
Although, according to the proof-of-concept nature of the present work, the proposed fractal dimension... present work is the low spatial resolution of the EEG system. A source localization study using a high-density EEG system (>= 64 channels) might help to elucidate which brain regions are mostly affected by hypoglycemia and exhibit both decrease in complexity and increase in delta and theta power [35]. Finally, as far as clinical applications, analysis of the EEG as alternative or, more likely, complementary to BG continuous sensors [36] [37] has been suggested [38].

line 287: "...results equal to..."

line 115: Define here the EU for euglycemia and HYPO for hypoglycemia (were not defined)

Figure 3: Substitute "topographic maps" with  "Box plots"

SECOND: RESULTS

Add a Table with statistical significance of each brain site for EU and HYPO.

I still cannot see two important things that the authors claim and if not true the paper is not worth publishing:

1) In all patient Box plots, the statistical difference between EU and HYPO for all measures largely disappear.  The authors cannot just talk themselves out of it (for example by saying that there is lot of variance across patients etc.).

2) From Figures 3 and 4 in one patient, I continue to see that it is more the frontal brain sites than the rest of the sites that exhibit larger differences between EU and HYPO. Authors claim the opposite with respect to the relevant sites!
